# Axonal injury signaling is restrained by a spared synaptic branch

**Laura J Smithson[1], Juliana L Zang[1], Lucas Junginger[1], Thomas J Waller[1], Lauren Reilly-Jankowiak[2], Sophia A Khan[2], Ye Li[3], Dawen Cai[3], Catherine A Collins[1,2]***

[1]Department of Molecular, Cellular and Developmental Biology, University of Michigan, Ann Arbor, United States; [2]Department of Neurosciences, Case Western Reserve University, Cleveland, United States; [3]Department of Cell and Developmental Biology, University of Michigan, Ann Arbor, United States

## eLife Assessment

This **important** study leverages the power of Drosophila genetics and sparsely-labeled neurons to propose an intriguing new model for neuronal injury signaling. The authors present **convincing** evidence to show that the somatic response to axonal injury can be suppressed if the injury is not complete, suggesting the presence of a new mode of injury 'integration.' While the underlying mechanism of this fascinating observation has yet to be determined, the phenomenon itself will be of broad significance in the field.

*For correspondence:
cxc1215@case.edu

**Abstract** The intrinsic ability of injured neurons to degenerate and regenerate their axons facilitates nervous system repair; however, this ability is not engaged in all neurons and injury locations. Here, we investigate the regulation of a conserved axonal injury response pathway with respect to the location of damage in branched motoneuron (MN) axons in *Drosophila* larvae. The dileucine zipper kinase (DLK; also known as MAP3K12 in mammals and Wallenda (Wnd) in *Drosophila*) is a key regulator of diverse responses to axonal injury. In three different populations of MNs, we observed the same striking result that Wnd/DLK signaling becomes activated only in response to injuries that remove all synaptic terminals. Injuries that spared even a small part of a synaptic terminal were insufficient to activate Wnd/DLK signaling, despite the presence of extensive axonal degeneration. The regulation of injury-induced Wnd/DLK signaling occurs independently of its previously known regulator, the Hiw/PHR ubiquitin ligase. We propose that Wnd/DLK signaling regulation is linked to the trafficking of a synapse-to-nucleus axonal cargo and that this mechanism enables neurons to respond to impairments in synaptic connectivity.

## Introduction

Repair of nervous system damage requires an ability of neurons to regenerate axons and synaptic connections. However, this ability is not universally induced following injury, most notably following spinal cord injury. While many studies have identified extrinsic factors that influence or inhibit axonal regeneration, several studies have noted that the location of damage can influence the intrinsic ability of the neuron to mount a transcriptional response to the damage (*Fernandes et al., 1999*; *Lorenzana et al., 2015*; *Mason et al., 2003*; *Wang et al., 2024*). Some studies have noted an effect of distance from the cell body for long axons in the spinal cord (*Fernandes et al., 1999*; *Mason et al., 2003*; *Wang et al., 2024*). Other studies have noted that the location of injury with respect to an axonal branching point also strongly influences the response (*Lorenzana et al., 2015*; *Wu et al., 2007*). Even

in a strongly inhibitory environment to regeneration, dorsal column sensory axons show robust axonal growth when injured proximal to their bifurcation in the spinal cord (*Lorenzana et al., 2015*).

Here, we investigate the regulation of a conserved axonal injury response pathway with respect to the location of axonal injury. The dileucine zipper kinase (DLK; known as MAP3K12 in mammals and Wallenda (Wnd) in *Drosophila*) is a key regulator of diverse responses to axonal injury. These include an essential role in the ability of damaged neurons to initiate axonal regeneration in worm and fly models (*Chen et al., 2011*; *Hammarlund et al., 2009*; *Stone et al., 2014*; *Xiong et al., 2010*; *Yan et al., 2009*), synaptic repair and recovery following CCR5 inhibition in a stroke model (*Joy et al., 2019*), and enhanced regeneration and mechanical allodynia following PNS nerve damage in mice (*Hu et al., 2019*; *Wlaschin et al., 2018*). Dichotomously, DLK is also required for the death of retinal ganglion cells following optic nerve damage (*Watkins et al., 2013*; *Welsbie et al., 2017*; *Welsbie et al., 2013*). In mammalian as well as in fly neurons, this kinase associates with vesicles that are physically transported in axons (*Holland et al., 2016*; *Xiong et al., 2010*), while downstream nuclear signaling requires functional axonal transport machinery (*Xiong et al., 2010*). DLK is therefore considered to function as a 'sensor' of axonal damage, whose activation can confer responses of repair or death, depending upon the cellular context (*Asghari Adib et al., 2018*).

While the responses gated by DLK are impactful for neurons and their circuits, the mechanism(s) that lead to DLK signaling activation are still poorly understood. A number of observations have documented DLK signaling activation in neurons that are not mechanically damaged but have experienced some form of cellular stress. These include the presence of cytoskeletal mutations (*Bounoutas et al., 2011*; *Chen et al., 2014*; *Kurup et al., 2015*; *Valakh et al., 2013*) and the presence of chemotherapy agents (*Bhattacharya et al., 2012*; *DeVault et al., 2024*; *Valakh et al., 2015*) known to impair axonal cytoskeleton integrity and transport. DLK activation is also responsible for phenotypes associated with mutations in the unc-104/KIF1A kinesin (*Li et al., 2017*), a major carrier of synaptic vesicle precursors in axons (*Guedes-Dias and Holzbaur, 2019*; *Petzoldt, 2023*). Inhibition of DLK is protective in mouse models of Amyotrophic Lateral Sclerosis (ALS) and Alzheimer's disease (*Le Pichon et al., 2017*; *Patel et al., 2017*). These observations have fostered growing interest in DLK as a potential therapeutic target and in understanding the mechanisms that control DLK signaling activation in neurons.

Here, we test the hypothesis that DLK/Wnd signaling is tuned to the synaptic connectivity of a neuron. A shared feature of nerve injuries and stressors that disrupt axonal cytoskeleton and transport is a loss in downstream connections. A conserved regulator of DLK/Wnd, the E3-ubiquitin ligase PAM/Highwire/Rpm-1/Phr1 (*Collins et al., 2006*; *Huntwork-Rodriguez et al., 2013*; *Nakata et al., 2005*), is hypothesized to function at synaptic terminals (*Abrams et al., 2008*; *Opperman and Grill, 2014*; *Xiong et al., 2012*; *Zhen et al., 2000*). This led us to ask whether interactions at an intact synaptic terminal are responsible for restraining Wnd signaling in uninjured neurons. We probed this hypothesis through injuries to branched motoneuron (MN) axons in *Drosophila* larvae, which allowed us to compare injuries that leave spared synaptic terminals to injuries that lead to complete denervation. In three different populations of MNs, we observed the same striking result that Wnd signaling becomes activated only in response to injuries that remove all synaptic terminals. Injuries that spared even a small part of a synaptic terminal did not activate Wnd signaling, despite the presence of extensive axonal degeneration. Surprisingly, removal of all synapses led to additive induction of Wnd signaling in *hiw* mutants. These observations suggest the existence of a mechanism that restrains Wnd signaling at synaptic terminals independently of the Hiw ubiquitin ligase.

## Results

### The presence of a spared synaptic branch restrains Wnd-mediated injury signaling in SNc MNs

To determine whether synaptic connections influence injury signaling by Wnd/DLK, we established methods to injure single synaptic branches of defined larval MNs. The m12 (5053A)-Gal4 driver line (*Ritzenthaler et al., 2000*) that we have used in previous nerve injury studies (*Xiong et al., 2010*; *Xiong and Collins, 2012*), drives expression in two single MNs that project closely fasciculated axons to innervate body wall muscles 26, 27, and 29 (*Figure 1A*). This pattern was previously attributed to an MN named MNSNc, which was noted to have 'two cell bodies' (*Kim et al., 2009*). We used the Bitbow2 (*Li et al., 2021*) multi-colored cell labeling approach to resolve the two neurons and

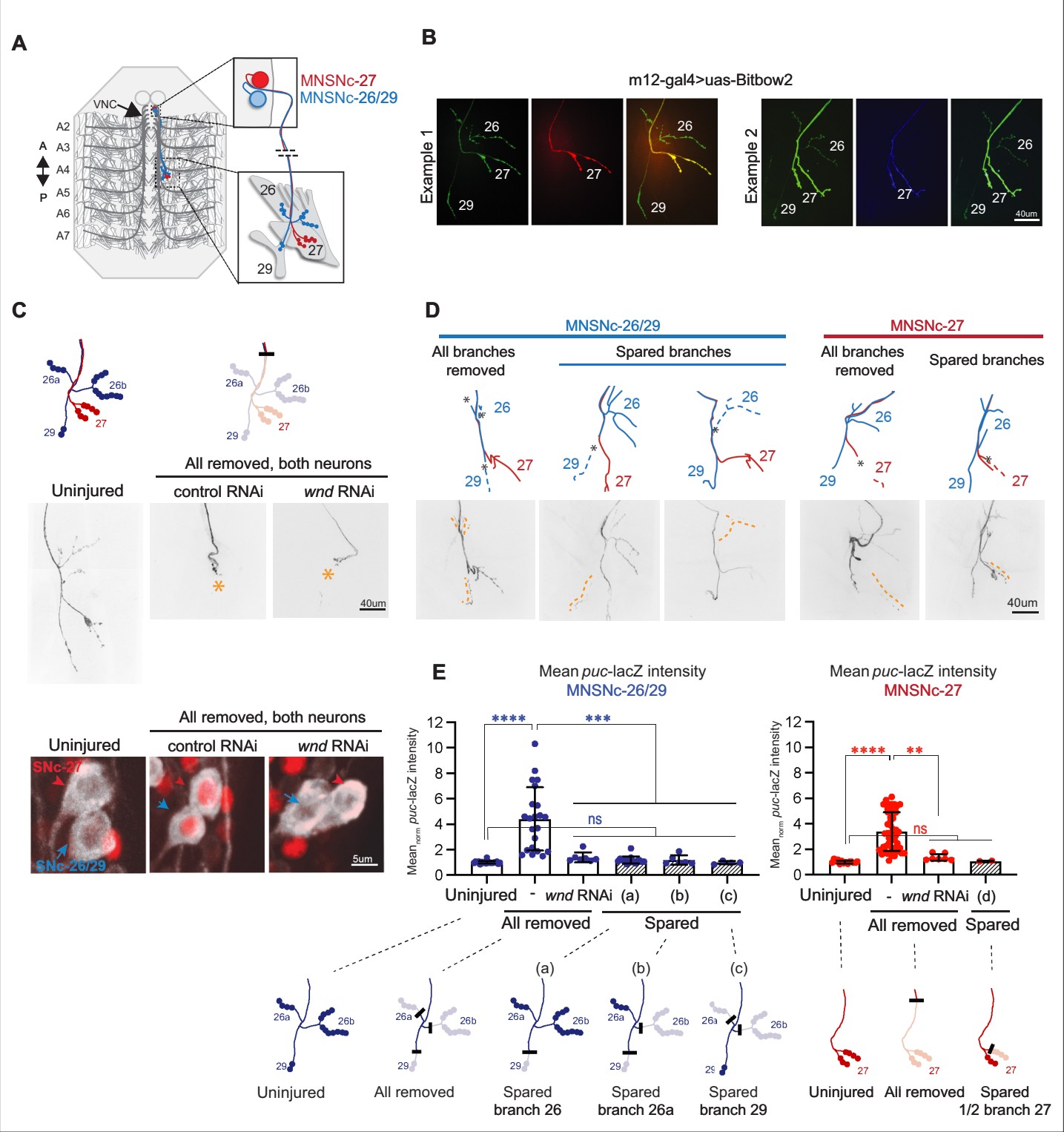

**Figure 1.** A spared synaptic branch restrains Wnd-dependent injury signaling in SNc motoneurons. (**A**) Schematic representation of the two SNc motoneurons innervating muscles 26 and 29 (MNSNc-26/29, red) and muscle 27 (MNSNc-27, blue), which are labeled by expression of the m12-Gal4 driver. (**B**) Example images of NMJ terminals from m12-Gal4/+; UAS-BitBow2 (*Li et al., 2021*)/+third instar larvae, used to define the connectivity shown in A. The neuron that innervates muscle 27 (MNSNc-27) expresses a distinct set of colors from the Bitbow2 (*Li et al., 2021*) reporter than the neuron that innervates muscles 26 and 29 (MNSNc-26/29). (**C**) Confirmation of *puc*-lacZ induction following laser axotomy. The cartoons on the top row show the location used to injure both axons; this location removes all of the synaptic branches from both MNSNc-26/29 and MNSNc-27. The middle row

*Figure 1 continued on next page*

*Figure 1 continued*

shows example injuries (versus uninjured, right) at the indicated location in m12-Gal4, UAS-mCD8GFP/*puc*-lacZ larvae. The bottom row shows examples of *puc*-lacZ expression (red channel) in the MNSNc cell bodies 24 hr following injury. (D) Example MNSNc-26/29 (blue) and MNSNc-27 (red) neurons injured at different locations. (E) Quantification of *puc*-lacZ intensity measurements in MNSNc-26/29 (blue) and MNSNc-27 (red) following injuries that remove all synaptic branches versus injuries that leave a spared synaptic branch. Injury location (**a**) removes the small number of boutons on muscle 29 while sparing the boutons on muscle 26. Injury location (**b**) removes boutons from muscle 29 and the posterior sub-branch on muscle 26. Injury location (**c**) removes all branches except for the small number of boutons on muscle 29. Note that all injuries that leave spared boutons (hatched shading) show no *puc*-lacZ induction, regardless of the number of boutons lost or spared. A one-way ANOVA with Tukey test for multiple comparisons was performed for each neuron. ****$p < 0.0001$; ***$p < 0.001$; **$p < 0.01$; ns = not significant.

The online version of this article includes the following figure supplement(s) for figure 1:

**Figure supplement 1.** Further characterization of branched injury assays to SNc and aCC motoneurons.

observed an invariable pattern that one MNSNc neuron innervates muscles 26 and 29, while the other innervates muscle 27. *Figure 1B* shows two examples: the neuron that innervates muscle 27 (middle images) expresses a set of colors that distinguish it from the terminals on muscles 26 and 29. We therefore refer to these two neurons labeled by the m12Gal4 driver as MNSNc-26/29 and MNSNc-27 (*Figure 1A, B*, *Figure 1—figure supplement 1*). MNSNc-26/29 (cartooned in blue in *Figure 1A*) has three collateral branches: two branches innervate muscle 26, and a single branch innervates muscle 29. MNSNc-27 (cartooned in red in *Figure 1A*) has two collateral branches that innervate muscle 27: these two branches most often remain together and occasionally bifurcate separately onto muscle 27 (*Figure 1—figure supplement 1A*). This innervation is stereotyped across segments and animals. As previously noted (*Kim et al., 2009*), the paired cell bodies of the MNSNc neurons are found on the lateral sides of the abdominal region of the ventral nerve cord (VNC). Bitbow2 expression revealed that the MNSNc-27 cell somas are positioned more ventrally compared to the cell somas of MNSNc-26/29 in the VNC (*Figure 1—figure supplement 1B*).

We used a pulsed dye laser to carry out axotomies of MNSNc axons at different locations in intact immobilized larvae (described in methods, *Smithson et al., 2025*). Successful injuries were confirmed by the degeneration of distal stumps within 24 hr post-injury. To assess the activation of Wnd signaling, we probed for the induction of *puckered* expression from the *puc*-lacZ enhancer trap (*Martin-Blanco et al., 1998*). Previous studies using this reporter line have shown that expression of nuclear localized lacZ is strongly induced following nerve injury, requiring Wnd, JNK, and the Fos transcription factor (*Xiong et al., 2010*). We confirmed this is the case for MNSNc neurons; axotomy of the m12-Gal4 labeled neurons upstream of the synaptic branches led to a fourfold induction of *puc*-lacZ expression in both neurons 24 hr after injury; this was abolished in neurons that co-express double-stranded RNAi targeting *wnd* (*Figure 1C*). Similar results were observed for nerve crush injuries (*Figure 1—figure supplement 1C*).

In contrast to axotomies that removed all synaptic branches, laser injury to single collateral branches of MNSNc-26/29, including branches innervating either muscle 26 or 29 and resulting in spared synaptic branches, did not induce *puc*-lacZ expression (*Figure 1D, E*). In addition, laser ablation of the anterior branch of MNSNc-27 was also insufficient to activate Wnd signaling (*Figure 1D, E*). We note that the induction of puc-lacZ did not correlate with the number of boutons that were lost or spared. MNSNc-26/29 forms fourfold more boutons on muscle 26 (17 ± 3.8) than 29 (4 ± 1.3). However, injuries that spared any of the branches, even the small number on muscle 29, showed equivalent puc-lacZ levels to uninjured neurons (*Figure 1E*; compare injury locations a, b, and c). We carried out similar experiments in aCC MNs, which can be labeled with the Dpr4-Gal4 driver (*Pérez-Moreno and O'Kane, 2019*). Laser axotomies that removed all of the synaptic branches resulted in a fourfold increase in *puc*-LacZ levels, while injuries that left spared synaptic boutons did not induce *puc*-lacZ expression (*Figure 1—figure supplement 1D, E*). These observations suggested that even a small number of remaining boutons was sufficient to restrain the activation of *puc*-lacZ expression.

Despite the small distance from the disconnected muscle, none of the injured MNSNc synaptic branches were able to re-innervate the muscle. We think this is due to an absence of axon growth-promoting cues, since MNSNc axons did show robust but misdirected axonal growth into the segmental nerve SNa following injuries in locations upstream of the synaptic branches (data not shown). However, we did notice differences in the trafficking of proteins to injured proximal stumps. An example of this is shown for ectopically expressed kinase-dead Wnd transgenic protein,

GFP-Wnd-KD in *Figure 1—figure supplement 1F, G*. We were only able to track kinase-inactive Wnd since overexpression of Wnd causes dramatic morphological defects to neurons (*Collins et al., 2006*; *Feoktistov and Herman, 2016*; *Xiong et al., 2010*). GFP-Wnd-KD was strongly induced and accumulated at the proximal stump following injuries that removed all synaptic branches but was barely detectable following injuries that left spared synaptic branches (*Figure 1—figure supplement 1F, G*). Collectively, these observations suggest that the presence of spared synaptic branches affects the subsequent events that occur in the injured axon. These include the stability and/or trafficking of Wnd protein in injured axons and the activation of Wnd-regulated signaling in the neuron soma.

## Restraint of Wnd-dependent injury signaling in bifurcated axons of type II VUM MNs

A more extreme example of axonal branching is illustrated by the ventral unpaired median neurons, which project symmetric bifurcated axons through separate nerves to innervate multiple body wall muscles on both left and right sides of the larva (*Figure 2A*, *Figure 2—figure supplement 1*; *Koon et al., 2011*; *Monastirioti et al., 1995*; *Vömel and Wegener, 2008*). VUM neurons can be specifically labeled based on their expression of tyrosine decarboxylase 2 using the Tdc2-Gal4 driver. Each abdominal segment has three VUM neurons, each of which sends a single axon dorsally which then bifurcates in the midline VNC (*Figure 2—figure supplement 1*; *Vömel and Wegener, 2008*). The two bifurcations then project through separate nerves to symmetrically innervate both left and right halves of the larval body. Through nerve crush injuries to only one ventral side of the animal, we were able to injure one bifurcation while leaving the other bifurcation intact. Successful injuries were determined based on the degeneration of the distal axon and synaptic terminals at 24 hr post-crush (*Figure 2B*). Whether both bifurcations, a single bifurcation, or neither bifurcation was injured was scored for each VUM neuron by tracing the Tdc2-Gal4, UAS-mCD8GFP labeled axons from the nerve to the cell body.

Similarly to other MNs (*Xiong et al., 2010*), *puc*-lacZ expression is barely detectable in uninjured VUM MNs and, compared to uninjured VUM neurons, is induced almost threefold at 24 hr following complete/full nerve crush injuries that damage both bifurcations and remove all synaptic connections (*Figure 2C, D*). This induction is abolished in VUM neurons that co-express *wnd*-RNAi (*Figure 2C, D*). Note that *wnd*-RNAi is only expressed in the VUM neurons and does not affect the other MNs which do not express Tdc2-Gal4. In contrast to full nerve crushes, injuries to nerves on a single side of the animal that the other bifurcation was intact ('half crush' injuries) did not induce *puc*-lacZ expression in VUM neurons (*Figure 2C, D*). Most MNs make ipsilateral and not bilateral projections; in 'half-crush' injuries, *puc*-lacZ is induced in most non-VUM MNs on the side of the crush, but is not induced in VUMs. These combined observations suggest that in *Drosophila* MNs, Wnd signaling is not tuned to detect axonal damage per se, but is instead uniquely tuned to detect a complete loss of innervation, which occurs following some injuries but not others.

## Restraint of Wnd signaling at spared branches does not require synaptic transmission

Since the presence of intact synaptic boutons restrains Wnd signaling activation, we considered whether cellular events associated with evoked or spontaneous synaptic transmission are associated with this mechanism. Summarized, we tested a total of 22 genetic manipulations expected to inhibit synaptic transmission, but none led to a change in *puc*-lacZ expression. These include electrical silencing of SNc MNs by Gal4/UAS-mediated expression of the *Drosophila* open rectifier K$^+$ channel (dORK) (*Nitabach et al., 2002*), silencing of transmission using temperature-sensitive mutations in dynamin (*Kitamoto, 2001*), and light-induced silencing of neurons expressing *Guillardia theta* anion channelrhodopsin 1 (gtACR1) (*Mohammad et al., 2017*; *Supplementary file 1*). Consistent with these negative results, we noted that previous studies have described many genetic manipulations that perturb evoked and/or spontaneous synaptic transmission (*Choi et al., 2014*; *Daniels et al., 2006*; *DiAntonio and Schwarz, 1994*; *Han et al., 2022*; *Kim et al., 2012*; *Mosca et al., 2005*; *Pittendrigh et al., 1997*) do not yield synaptic phenotypes (of synaptic overgrowth or decreased VGlut expression levels) associated with Wnd activation (*Collins et al., 2006*; *Li et al., 2017*).

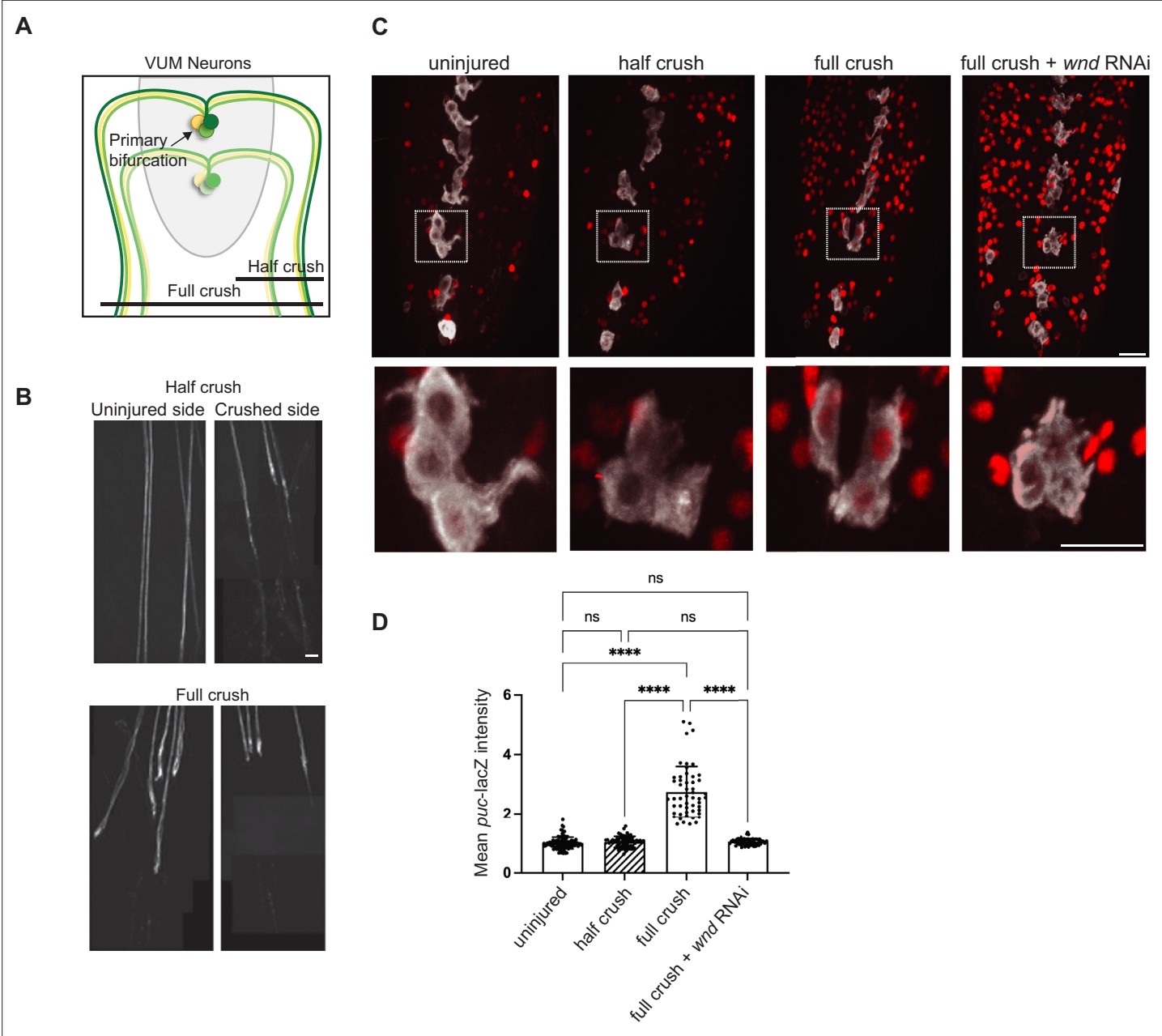

**Figure 2.** Restraint of Wnd-mediated injury signaling by spared branch in bifurcated neurons. (**A**) Cartoon of ventral unpaired median (VUM) neurons, which have bifurcated axons that symmetrically innervate body wall muscles on both the left and right sides of the animal. Nerve crush to either left or right side of the animal can axotomize a single bifurcation while leaving the other bifurcated axon intact. (**B**) Example images of VUM axons (visualized in Tdc2-Gal4, UAS-mCD8-GFP larvae) in segmental nerves on the uninjured and injured sides following nerve crush to a single side. (**C**) Example images of *puc*-lacZ expression in the VNC (ventral nerve cord) of larvae following nerve crush to a single side (half crush) versus crush to all the segmental nerves (full crush). *puc*-lacZ expression (red) is induced in VUM neurons (white) only after full crush. In contrast, other motoneurons, which innervate a single side, are induced by both half and full crush injuries. Co-expression of UAS-*wnd*-RNAi in VUM neurons cell autonomously inhibits *puc*-lacZ induction. (**D**) Quantification of *puc*-lacZ intensity measurements in VUM neurons. A one-way ANOVA with Tukey test for multiple comparisons was performed. ****$p < 0.0001$; ns = not significant. Scale bars = 20 μm.

The online version of this article includes the following figure supplement(s) for figure 2:

**Figure supplement 1.** Anatomy and laser surgery of Tdc2 bifurcated neurons.

## Restraint of Wnd-mediated axon injury signaling is independent of the Highwire ubiquitin ligase

We then asked whether a known upstream regulator of Wnd signaling, the Pam/Hiw/Rpm-1 (PHR) ubiquitin ligase, functions to restrain Wnd at synaptic branches. PHR is a large protein with multiple evolutionarily conserved domains, inducing a RING-finger domain, which regulates DLK/Wnd in invertebrate (*C. elegans* and *Drosophila*) (*Collins et al., 2006*; *Nakata et al., 2005*) and vertebrate (*Huntwork-Rodriguez et al., 2013*) model organisms. PHR is an attractive candidate since it localizes to synaptic terminals (*Abrams et al., 2008*; *Opperman and Grill, 2014*; *Zhen et al., 2000*) and loss of PHR function leads to increased levels of Wnd/DLK at synapses (*Collins et al., 2006*; *Nakata et al., 2005*). This hypothesis predicts that removal of all synaptic branches would be equivalent to a genetic loss in PHR function. We tested this in null mutants for the *Drosophila* ortholog of PHR, Highwire (Hiw). The *hiw*[ΔN] mutation deletes the entire N-terminal half of the *hiw* gene and abolishes expression of the Hiw protein (*Wu et al., 2005*). Since *hiw*[ΔN] animals are viable, we were able to carry out injury assays in neurons that completely lack Hiw function.

Consistent with previously reported phenotypes for *hiw* in other MN types (*Collins et al., 2006*; *Wan et al., 2000*; *Wu et al., 2005*), uninjured MNSNc neurons in male *hiw*[ΔN] mutants have an increased number of axon collateral and terminal branches at muscles 26, 29, and 27 NMJs (*Figure 3A*, *Figure 3—figure supplement 1*). Also consistent with previous observations, uninjured neurons show elevated expression of *puc*-lacZ in *hiw*[ΔN] mutants compared to control animals (*Figure 3*, *Figure 3—figure supplement 1*). Strikingly, injuries that removed all synaptic terminals led to an even further elevation of *puc*-lacZ expression in *hiw*[ΔN] mutant neurons. This was the case for laser axotomies that removed all synaptic branches from either MNSNc-26/29 and/or MNSNc-27 neurons (*Figure 3A*, right column, and *Figure 3B*), and also for VUM neurons following nerve crush injuries (*Figure 3C*). Injuries to a single synaptic branch (on muscle 29) of MNSNc-26/29 in *hiw*[ΔN] mutants had a similar level of *puc*-lacZ expression as uninjured *hiw*[ΔN] neurons (*Figure 3A, B*). Collectively, these observations suggest that the presence of a spared synapse is capable of restraining Wnd signaling independently of Hiw's function.

## Discussion

### Axonal branches and spared synaptic connections influence the ability of injured axons to regenerate

Since the foundational observations of Ramon y Cajal, we have known that the ability of axons to regrow following damage varies strongly according to the location of damage (*Ramón y Cajal, 1928*). Most widely considered are differences in extrinsic factors that influence the ability of axons to grow following PNS injuries, and the impediments to axonal regeneration in the CNS that inhibit repair following spinal cord injuries (*Huebner and Strittmatter, 2009*). A less well-studied but important intrinsic determinant of regeneration ability is the location of the injury with respect to axonal branches. For *C. elegans* PLM mechanosensory neurons, robust regeneration occurs following injuries that remove both the synaptic and sensory branches, but not following injuries that leave a synaptic branch intact (*Wu et al., 2007*). In the mammalian spinal cord, an elegant study following responses to laser-induced microsurgery to ascending and descending central projections of sensory neurons observed that remarkable regeneration occurred following injuries proximal to the branching point (which led to removal of all branches) but not distal (which left one branch intact) (*Lorenzana et al., 2015*). Studies in mice have also implicated synaptic proteins alpha2-delta, Munc13, and RIM in restraining regenerative ability of axons (*Hilton et al., 2022*; *Tedeschi et al., 2016*). These observations are consistent with the possibility that spared afferent synaptic connections that remain following injuries distal to the branch point inhibit the regeneration ability of centrally projecting axons in the spinal cord.

It is noteworthy that many of the axons that project over great distances in the spinal cord have at least one synaptic branch. Neurons that project through the corticospinal tract (CST), whose poor regeneration ability following spinal cord injury is most widely studied, form synaptic branches in the red nucleus, brainstem, and throughout the spinal cord (*Raineteau and Schwab, 2001*). A recent study has profiled the responses of CST neurons at different injury locations (*Wang et al., 2024*). Strikingly, regeneration-associated genes (RAGs) and phosphorylated cJun, a marker of DLK signaling activation, are not induced in CST neurons following spinal cord injury but are robustly induced following

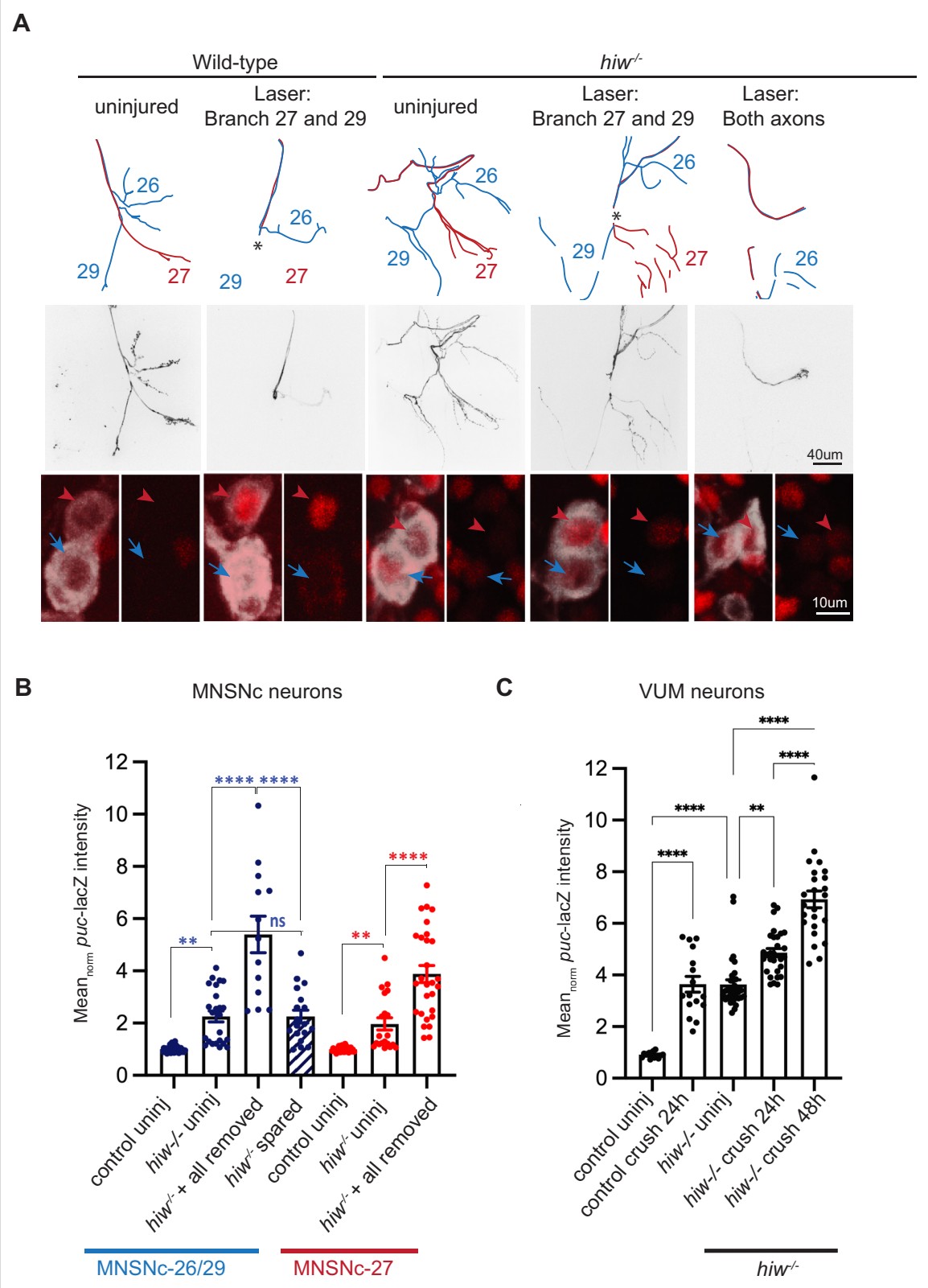

**Figure 3.** Presence of spared synaptic branch restrains Wnd signaling independently of Hiw. (**A**) Laser axotomy is carried out to MNSNc neurons at a location (indicated by asterisk (*)) that completely removes the synaptic terminal of MNSNc-27 (red neuron). The injury also leads to loss of the MNSNc-26/29 (blue) terminal on muscle 29 but not 26, hence leaves a spared synaptic branch. The final column shows an axotomy that fully removes the terminals for both MNSNc neurons. These injuries were repeated in control animals versus the background of a *hiw* null mutant, *hiw^ΔN*. (**B**) Quantification

*Figure 3 continued on next page*

*Figure 3 continued*

of *puc*-lacZ expression for individual MNSNc neurons after full versus spared axotomies, compared to uninjured neurons. Basal *puc*-lacZ expression is already elevated in uninjured *hiw*$^{AN}$ neurons compared to control. This can be further elevated in axotomies that remove all synapses, but not in axotomies that leave spared branches. (**C**) Quantification of *puc*-lacZ in VUM neurons (labeled by Tdc-2-Gal4; UAS-mCD8-GFP) 24 and 48 hr following full nerve crush in control versus *hiw*$^{AN}$ mutants. A two-way ANOVA with Tukey test for multiple comparisons was performed. ****$p < 0.0001$; ***$p < 0.001$; **$p < 0.01$; ns = not significant.

The online version of this article includes the following figure supplement(s) for figure 3:

**Figure supplement 1.** Confirmation that elevated *puc*-lacZ expression in *hiw* mutants requires Wnd.

intracortical injuries close to CST cell bodies (*Mason et al., 2003*; *Wang et al., 2024*). Since the latter may be the only form of injury that removes all synaptic branches from the CST neurons, we propose that restraint of DLK signaling activation by spared synaptic branches could be a prominent feature of the poor intrinsic regeneration capacity of neurons following spinal cord injury.

## Restraint of Wnd/DLK signaling at synaptic terminals

Using genetic manipulations that inhibit or perturb synaptic transmission and/or neuronal excitability, we did not detect a requirement for synaptic transmission in the restraint of Wnd by spared synaptic branches. The PHR ubiquitin ligase, known as Hiw in *Drosophila*, was a logical candidate to regulate Wnd at synapses, since studies in multiple model organisms have shown that loss of this enzyme leads to increased levels of Wnd/DLK in axons (*Collins et al., 2006*; *Huntwork-Rodriguez et al., 2013*; *Nakata et al., 2005*). The *puc*-lacZ reporter implies that Wnd signaling is elevated in *hiw* mutants. However, the restraint conferred by spared synaptic branches is still active in the absence of Hiw, since Wnd signaling can be further elevated by axotomy of all synapses in *hiw* null mutants (*Figure 3*). In a previous study, we noted that the nerve crush injury led to a rapid down-regulation of an ectopically expressed Hiw-GFP transgene and speculated that impairment of restraint by Hiw leads to activation of Wnd signaling in injured axons (*Xiong et al., 2010*). Our current data do not rule out a role for Hiw but suggest the existence of additional mechanisms that restrain Wnd signaling at intact synapses. This has also been suggested from developmental studies of photoreceptor growth cone termination, in which Hiw-independent downregulation of Wnd protein occurs concomitantly with the development of presynaptic boutons (*Feoktistov and Herman, 2016*).

We speculate that the regulation of Wnd is linked to the trafficking of organelles between the synaptic terminal and cell body, akin to neurotrophin signaling, which relies on retrograde trafficking of signaling endosomes in axons (*Cosker et al., 2008*). Consistent with this idea, DLK signaling becomes activated following nerve growth factor withdrawal from distal axons (*Ghosh et al., 2011*; *Larhammar et al., 2017*). Previous studies of Wnd signaling have documented its dependence on retrograde axonal transport machinery (*Xiong et al., 2010*). Moreover, mutations that disrupt axonal cytoskeleton and the unc-104/kif1A kinesin also lead to Wnd/DLK signaling activation (*Li et al., 2017*; *Valakh et al., 2015*; *Valakh et al., 2013*). Both Wnd and its homologue DLK in mice show regulated association with organelle membranes via palmitoylation, and disruption of its palmitoylation abolishes DLK's signaling ability (*Holland et al., 2016*; *Kim et al., 2024*; *Niu et al., 2022*). Palmitoylation and depalmitoylation are dynamically regulated in axons (*Ramzan et al., 2023*; *Zhang et al., 2024*), hence comprise an attractive mechanism for mediating restraint of DLK at synaptic branches. Future delineation of the organelle(s) that Wnd/DLK associates with may provide important clues to its mechanism of regulation.

Our observations that Wnd signaling could be restrained by an intact axonal bifurcation suggest that at least one level of regulation could occur in the cell body. Consistent with this idea, a recent study has shown that Wnd signaling can be ectopically activated in the cell body when its transport to distal synapses is impaired in *rab11* mutants (*Kim et al., 2024*). Given the many factors that have been documented thus far that regulate DLK/Wnd protein or signaling (*Asghari Adib et al., 2018*), we think that this kinase must be tightly regulated both at synapses and cell bodies (*Figure 4*). Regulation at both locations gives the neuron a way to monitor the state of its entire axon and restrict signaling activation to scenarios where all efferent connections of the axon are disrupted.

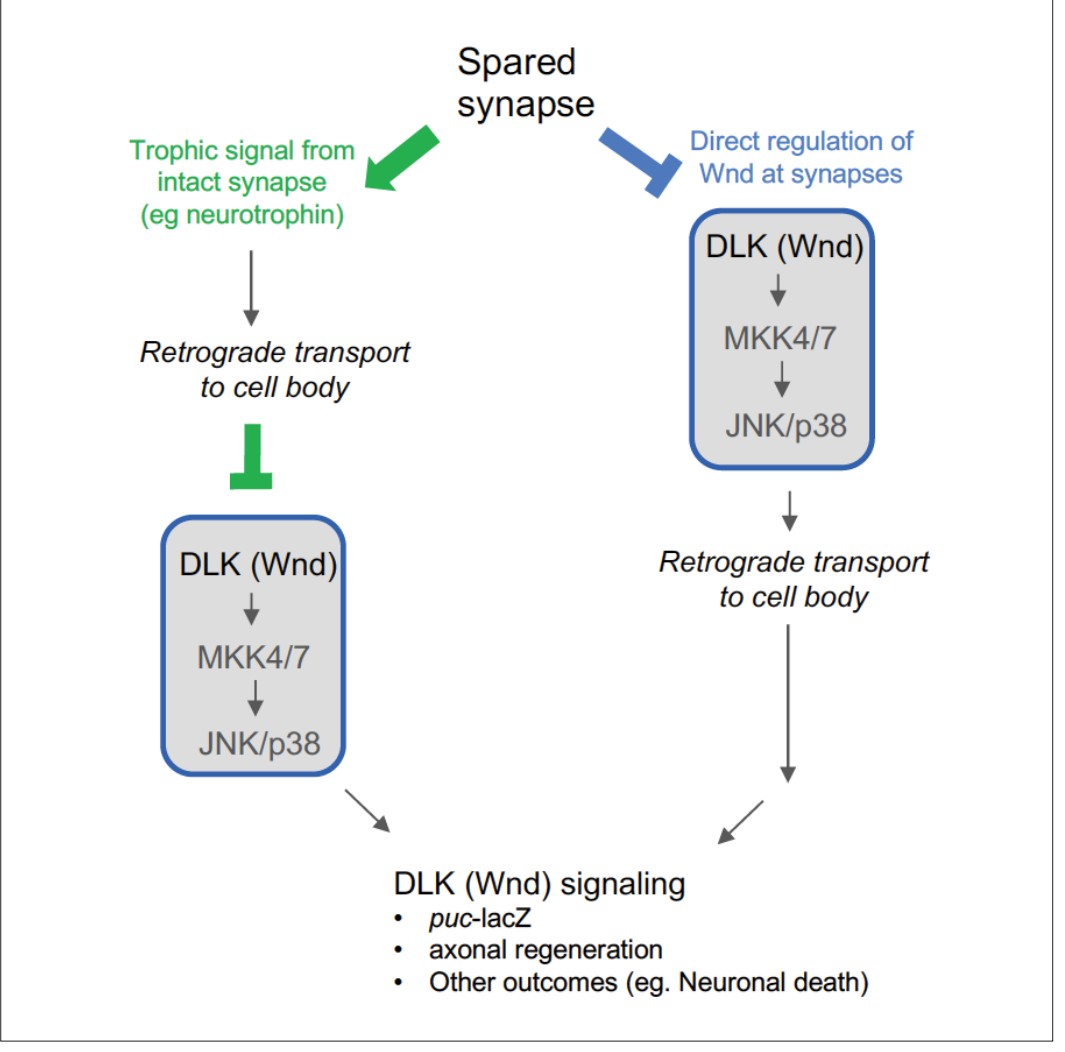

**Figure 4.** Potential mechanisms for regulation of Wnd signaling from synaptic terminals. In green, Wnd signaling may be regulated in the cell body downstream of a retrogradely transported signal (e.g., neurotrophin signaling). In blue, Wnd signaling activation is restrained locally at synaptic terminals, perhaps by regulating the levels or activation of Wnd itself. Activated Wnd or a downstream signaling factor is then retrogradely transported to the cell body. Previous observations that inhibition of retrograde transport blocks the induction of Wnd signaling following axonal injury favor the latter (blue) possibility. However, the restraint conferred by a physically separate bifurcation suggests that an inhibitor of Wnd signaling activation can be retrogradely transported (green). We speculate that both mechanisms act as dual checkpoints to restrain Wnd signaling activation in the context of healthy circuits.

## Materials and methods

**Key resources table**

| Reagent type (species) or resource | Designation | Source or reference | Identifiers | Additional information |
|---|---|---|---|---|
| Gene (*Drosophila melanogaster*) | *wnd* (*wallenda*) | Flybase | FBgn0036896 | |
| Gene (*Drosophila melanogaster*) | *puc* (*puckered*) | Flybase | FBgn0243512 | |
| Gene (*Drosophila melanogaster*) | *hiw* (*highwire*) | Flybase | FBgn0030600 | |

*Continued on next page*

*Continued*

| Reagent type (species) or resource | Designation | Source or reference | Identifiers | Additional information |
|---|---|---|---|---|
| Genetic reagent (*Drosophila melanogaster*) | UAS-mCD8-GFP | Bloomington *Drosophila* Stock Center (BDSC) | RRID:BDSC_5137 | *Lee and Luo, 1999* |
| Genetic reagent (*Drosophila melanogaster*) | m12-gal4 (P(Gal4)$^{5053A}$) | Bloomington *Drosophila* Stock Center (BDSC) | RRID:BDSC_2702 | *Ritzenthaler et al., 2000* |
| Genetic reagent (*Drosophila melanogaster*) | BG380-Gal4 | Bloomington *Drosophila* Stock Center (BDSC) | RRID:BDSC_42736 | *Budnik et al., 1996*; *Sanyal, 2009* |
| Genetic reagent (*Drosophila melanogaster*) | *puc*-lacZ[E69] | Bloomington *Drosophila* Stock Center (BDSC) | RRID:BDSC_98329 | *Martin-Blanco et al., 1998* |
| Genetic reagent (*Drosophila melanogaster*) | *puc*-GFP | Melissa Rolls | | *Rao and Rolls, 2017* |
| Genetic reagent (*Drosophila melanogaster*) | *hiw$^{AN}$* | Bloomington *Drosophila* Stock Center (BDSC) | RRID:BDSC_51637 | *Wu et al., 2005* |
| Genetic reagent (*Drosophila melanogaster*) | UAS-Bitbow2 | Dawen Cai | | *Li et al., 2021* |
| Genetic reagent (*Drosophila melanogaster*) | Tdc2-Gal4 | Bloomington *Drosophila* Stock Center (BDSC) | RRID:BDSC_9313 | |
| Genetic reagent (*Drosophila melanogaster*) | UAS-*wnd*-RNAi | Bloomington *Drosophila* Stock Center (BDSC) | RRID:BDSC_35369 | |
| Genetic reagent (*Drosophila melanogaster*) | UAS-*lexA*-RNAi | Bloomington *Drosophila* Stock Center (BDSC) | RRID:BDSC_67947 | |
| Genetic reagent (*Drosophila melanogaster*) | Additional *Drosophila* lines are summarized in **Supplementary file 1** | | | |
| Antibody | Mouse monoclonal anti-lacZ | DSHB Cat# 40-1a | RRID:AB_528100 | 1:100 dilution |
| Antibody | Rabbit polyclonal anti-DsRed | Takara Bio Cat# 632496 | RRID:AB_10013483 | 1:1000 dilution |
| Antibody | A488 rabbit polyclonal anti-GFP | Molecular Probes Cat# A-21311 | RRID:AB_221477 | 1:1000 dilution |
| Antibody | AlexFluor 568 goat polyclonal anti-mouse | Thermo Fisher, A11004 | RRID:AB_2534072 | 1:1000 dilution |
| Antibody | AlexFluor 488 goat polyclonal anti-mouse | Thermo Fisher A32723 | RRID:AB_2633275 | 1:1000 dilution |
| Antibody | AlexFluor 568 goat polyclonal anti-rabbit | Thermo Fisher A-11011 | RRID:AB_143157 | 1:1000 dilution |
| Chemical compound, drug | Paraformaldehyde Aqueous Solution EM Grade | Electron Microscopy Sciences | Cat #15710 | 16% aqueous solution diluted to 4% in PBS. Used within 1 week of dilution. |
| Biological sample (goat) | Normal goat serum (NGS) | Fisher Scientific | Cat #16210064 | Diluted to 5% in PBS |
| Chemical compound, drug | Prolong Diamond Antifade media | Thermo Fisher Scientific, P36970 | Cat #P36970 | |

*Continued on next page*

*Continued*

| Reagent type (species) or resource | Designation | Source or reference | Identifiers | Additional information |
|---|---|---|---|---|
| Other | PDMS microfluidic chip for immobilizing larvae | MicroKosmos | 'Mechanical Immobilization Chip' for *Drosophila* larvae | (specialized equipment) Larva chip is available for purchase from https://www.ukosmos.com |
| Other | Dumont #5 fine forceps | Roboz Surgical | Cat #RS4978 | (specialized equipment) Forceps for carrying out the peripheral nerve crush assay |
| Software, algorithm | Volocity software 6.2 | Improvision, PerkinElmer | | |
| Software, algorithm | GraphPad Prism | | | |

### *Drosophila* stocks/genetics

All fly crosses were raised at 25°C in a 12-hr light/dark cycle on standard sucrose and yeast food. Both male and female larvae were used unless otherwise stated. Strains used are listed above in the Key Resources Table. Additional stocks are listed in *Supplementary file 1*.

### Immunohistochemistry

Wandering second and third instar larvae were dissected in ice-cold 1x PBS, fixed in 4% paraformaldehyde (16% diluted in 1X PBS, Electron microscopy Labs) for 20 min at room temperature and washed thrice in 1X PBS. Tissues were blocked for a minimum of 30 min in 5% normal goat serum (NGS) in 0.25% Triton X-100 in 1X PBS (PBST). Primary antibodies were incubated overnight in 5% NGS in 0.25% PBST at room temperature unless otherwise stated. Tissues were washed thrice in 1x PBS and mounted onto superfrost plus slides (Fisher Scientific) and coverslipped with ProLong Diamond Antifade media (Thermo Fisher Scientific, P36970). Antibodies and concentrations are listed in the Key Resources Table. All experimental and control larval groups were processed (dissected, fixed, stained, and images captured) together using identical confocal settings.

### Nerve crush assays

Peripheral nerve crushes in wandering second instar larvae were performed as previously described (*Waller et al., 2025*; *Xiong et al., 2010*). Briefly, with the ventral side facing upwards, anesthetized larvae (taken from 1X PBS and placed on $CO_2$ pad for ~3–5 min) had a small region of the cuticle (around abdominal segment 2) with underlying segmental nerves gently pinched with a pair of Dumont #5 fine forceps (Roboz Surgical RS4978). After injury, larvae were placed in small dishes containing standard fly food and kept at 25°C (12 hr light/dark) for 24 hr. Injuries were confirmed by the presence of posterior tail paralysis. For the new spared nerve crush assay, we performed injuries on wandering 2nd instar larvae with labeled type II octopaminergic neurons driven by Tdc2-gal4. The Tdc2-gal4 drives expression in three ventrally located midline neurons that send one axon to the left side and one axon to the right side of larvae. Similarly to complete nerve crushes, anesthetized larvae (ventral side up) were gently pinched with fine forceps on the left side (around abdominal segment 2), injuring nerves on one side while sparing nerves on the other side. Animals were placed in small dishes containing standard fly food and kept at 25°C (12 hr light/dark) for 24–72 hr.

### Axon/synaptic branch laser injury

As previously described (*Ghannad-Rezaie et al., 2012*; *Mishra et al., 2014*; *Smithson et al., 2024*; *Waller et al., 2025*), single wandering second instar larva were taken from a dish containing 1X PBS and gently placed onto a kim wipe to remove excess PBS and then dipped into halocarbon oil 700 (H8898, Sigma). Each larva was then placed onto a glass coverslip dorsal side up (for inverted confocal) with an anterior to posterior (left to right) orientation. A PDMS microfluidic chip (https://www.ukosmos.com/) was placed on top of the larvae applying light force. A tight seal was created by applying gentle suction from a 30-ml syringe plunger. This vacuum suction immobilizes the intact larvae. The coverslip containing the microfluidic chip and larvae was mounted onto an Improvision spinning disk confocal system (PerkinElmer) connected to a Micropoint Laser Illumination and Ablation System (Andor Technology). The method for laser-induced microsurgery is described in *Smithson et al., 2025*. Prior to injury, the laser strength and region of interest were calibrated and optimized.

Individual axonal branches (membrane GFP labeled) were identified and laser ablated. Confirmation of injury was demonstrated by a small absence of membrane GFP and degeneration of the proximal axon was detected as early as 8 hr after laser injury.

## Imaging, quantification, and analysis

Both experimental and control larvae were imaged on an Improvision spinning disk confocal system (PerkinElmer) and quantified using identical confocal settings by an independent experimenter blinded to the genotype/experiment. Injured m12-gal4 and Tdc2-gal4 axons were traced back to identify corresponding cell bodies in the lateral and medial VNC. Only the neurons whose identity and injury location could be clearly identified were used for analysis. To quantify *puc*-lacZ levels, the mean intensity of lacZ immunolabeling was measured in specific injury-confirmed neurons. *Puc*-lacZ contains an NLS sequence fused to lacZ, resulting in nuclear expression of *puc*. The mean lacZ intensity was measured per neuron and normalized to uninjured control neurons. All imaging, image contrast/color adjustments, and quantification were conducted using Volocity software 6.2 (Improvision, Perkin-Elmer). GraphPad Prism software was used for both selection and calculation of appropriate statistical tests. The methods used and *n* for each figure are reported in individual figure legends. p values are reported for 95% confidence intervals, and graphs are plotted with mean ± SD (standard deviation).

## Acknowledgements

We would like to thank Mariana Jimenez, Luis Rivera, and Sarah Cooke for help with tissue processing, Eric Robertson and Chris Jasinski for technical assistance. We thank Heather Broihier, Dion Dickman, Dhananjay Yellajoshyula, and Jerry Silver for helpful discussions and comments on the manuscript. We thank Yuanquan Song (University of Pennsylvania) for sharing piezo flies, Yujia (Henry) Hu and Bing Ye (University of Michigan) for gtACR1 flies and technical advice. Stocks obtained from the Bloomington *Drosophila* Stock Center (NIH P40OD018537) and Vienna *Drosophila* Resource Center (VDRC, http://www.vdrc.at; *Dietzl et al., 2007*) were used in this study. This research was funded by the National Institutes for Health (NS069844 to CAC) and by the Canadian Institutes for Health Research (CIHR Fellowship to LJS).

## Additional information

### Funding

| Funder | Grant reference number | Author |
|---|---|---|
| National Institutes of Health | NS069844 | Catherine A Collins |
| Canadian Institutes of Health Research | | Laura J Smithson |

The funders had no role in study design, data collection, and interpretation, or the decision to submit the work for publication.

### Author contributions

Laura J Smithson, Conceptualization, Data curation, Formal analysis, Funding acquisition, Validation, Investigation, Methodology, Writing – review and editing; Juliana L Zang, Conceptualization, Formal analysis, Investigation, Visualization, Methodology, Writing – review and editing; Lucas Junginger, Conceptualization, Investigation, Methodology; Thomas J Waller, Data curation, Validation, Investigation, Methodology, Writing – review and editing; Lauren Reilly-Jankowiak, Formal analysis, Validation, Investigation, Methodology; Sophia A Khan, Formal analysis, Validation, Methodology; Ye Li, Dawen Cai, Resources, Methodology; Catherine A Collins, Conceptualization, Data curation, Supervision, Funding acquisition, Writing - original draft, Project administration, Writing – review and editing

### Author ORCIDs

Laura J Smithson https://orcid.org/0000-0003-1529-3005
Juliana L Zang https://orcid.org/0000-0002-5738-8355

Thomas J Waller ⑩ https://orcid.org/0000-0002-3098-9070
Lauren Reilly-Jankowiak ⑩ https://orcid.org/0000-0002-7496-6866
Sophia A Khan ⑩ https://orcid.org/0009-0004-1416-5022
Ye Li ⑩ https://orcid.org/0000-0002-8647-384X
Dawen Cai ⑩ https://orcid.org/0000-0003-4471-2061
Catherine A Collins ⑩ https://orcid.org/0000-0002-1608-6692

Reviewer #1 (Public review): https://doi.org/10.7554/eLife.104896.3.sa1
Reviewer #2 (Public review): https://doi.org/10.7554/eLife.104896.3.sa2
Reviewer #3 (Public review): https://doi.org/10.7554/eLife.104896.3.sa3
Author response https://doi.org/10.7554/eLife.104896.3.sa4

## Additional files

### Supplementary files
MDAR checklist

Supplementary file 1. Summary of negative results from genetic manipulations that impair synaptic transmission and/or signaling at synapses.

### Data availability
Confocal images and quantification for key data in Figure 1 have been deposited on Dryad. Dataset DOI: https://doi.org/10.5061/dryad.wm37pvn1k.

The following dataset was generated:

| Author(s) | Year | Dataset title | Dataset URL | Database and Identifier |
|---|---|---|---|---|
| Smithson LJ | 2025 | Images and quantification of puc-lacZ intensity to assess Wnd signaling activation | https://doi.org/10.5061/dryad.wm37pvn1k | Dryad Digital Repository, 10.5061/dryad.wm37pvn1k |

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
