## [Editor Report · eLife Assessment]

This **important** study leverages the power of Drosophila genetics and sparsely-labeled neurons to propose an intriguing new model for neuronal injury signaling. The authors present **convincing** evidence to show that the somatic response to axonal injury can be suppressed if the injury is not complete, suggesting the presence of a new mode of injury 'integration.' While the underlying mechanism of this fascinating observation has yet to be determined, the phenomenon itself will be of broad significance in the field.

---

## [Referee Report · Reviewer #1 (Public review)]

This manuscript presents an interesting exploration of the potential activation mechanisms of DLK following axonal injury. While the experiments are beautifully conducted and the data are solid, I feel that there is insufficient evidence to fully support the conclusions made by the authors.

In this manuscript, the authors exclusively use the puc-lacZ reporter to determine the activation of DLK. This reporter has been shown to be induced when DLK is activated. However, there is insufficient evidence to confirm that the absence of reporter activation necessarily indicates that DLK is inactive. As with many MAP kinase pathways, the DLK pathway can be locally or globally activated in neurons, and the level of DLK activation may depend on the strength of the stimulation. This reporter might only reflect strong DLK activation and may not be turned on if DLK is weakly activated.

As noted by the authors, DLK has been implicated in both axon regeneration and degeneration. Following axotomy, DLK activation can lead to the degeneration of distal axons, where synapses are located. This raises an important question: how is DLK activated in distal axons? The authors might consider discussing the significance of this "synapse connection-dependent" DLK activation in the broader context of DLK function and activation mechanisms.

---

## [Referee Report · Reviewer #2 (Public review)]

Summary:

The authors study a panel of sparsely labeled neuronal lines in Drosophila that each form multiple synapses. Critically, each axonal branch can be injured without affecting the others, allowing the authors to differentiate between injuries that affect all axonal branches versus those that do not, creating spared branches. This is a highly powerful model. Axonal injuries are known to cause Wnd (mammalian DLK)-dependent retrograde signals to the cell body, culminating in a transcriptional response. This work identifies a fascinating new phenomenon that this injury response is not all-or-none. If even a single branch remains uninjured, the injury signal is not activated in the cell body. The authors rule out that this could be due to changes in the abundance of Wnd (perhaps if incrementally activated at each injured branch) by Wnd, Hiw's known negative regulator. Thus there is both a yet-undiscovered mechanism to regulate Wnd signaling, and more broadly a mechanism by which the neuron can integrate the degree of injury it has sustained. It will now be important to tease apart the mechanism(s) of this fascinating phenomenon. But even absent a clear mechanism, this is a new biology that will inform the interpretation of injury signaling studies across species.

Strengths:

- A conceptually beautiful series of experiments that reveal a fascinating new phenomenon is described, with clear implications (as the authors discuss in their Discussion) for injury signaling in mammals.

- Suggests a new mode of Wnd regulation, independent of Hiw.

Weaknesses:

-The use of a somatic transcriptional reporter for Wnd activity is powerful, however, the reporter indicates whether the transcriptional response was activated, not whether the injury signal was received. It remains possible that Wnd is still activated in the case of a spared branch, but that this activation is either local within the axons (impossible determine in the absence of a local reporter) or that the retrograde signal was indeed generated but it was somehow insufficient to activate transcription when it entered the cell body. This is more of a mechanistic detail (and likely an extreme technical challenge to assess) and should not detract from the overall importance of the study

-That the protective effect of a spared branch is independent of Hiw, the known negative regulator of Wnd, is fascinating. But this leaves open a key question: what is the signal?

Comments on revisions:

I appreciate your discussion about the potential bi-modal regulation of the puckered transcriptional reporter and think that readers would benefit from a short discussion of this.

---

## [Referee Report · Reviewer #3 (Public review)]

Summary:

This manuscript seeks to understand how nerve injury-induced signaling to the nucleus is influenced, and it establishes a new location where these principles can be studied. By identifying and mapping specific bifurcated neuronal innervations in the Drosophila larvae, and using laser axotomy to localize the injury, the authors find that sparing a branch of a complex muscular innervation is enough to impair Wallenda-puc (analogous to DLK-JNK-cJun) signaling that is known to promote regeneration. It is only when all connections to the target are disconnected that cJun-transcriptional activation occurs.

Overall, this is a thorough and well-performed investigation of the mechanism of spared-branch influence on axon injury signaling. The findings on control of wnd are important because this is a very widely used injury signaling pathway across species and injury models. The authors present detailed and carefully executed experiments to support their conclusions. Their effort to identify the control mechanism is admirable and will be of aid to the field as they continue to try to understand how to promote better regeneration of axons.

Strengths:

The paper does a very comprehensive job of investigating this phenomenon at multiple locations and through both pinpoint laser injury as well as larger crush models. They identify a non-hiw based restraint mechanism of the wnd-puc signaling axis that presumably is originating from the spared terminal. They also present a large list of tests they performed to identify the actual restraint mechanism from the spared branch, which has ruled out many of the most likely explanations. This is an extremely important set of information to report, to guide future investigators in this and other model organisms on mechanisms by which regeneration signaling is controlled (or not).

Weaknesses:

While there are many questions raised by these results that are not answered here, including the pathways upstream and downstream of DLK and how the binary switch control of DLK/puc signaling is executed, the model built in this manuscript is valuable to future work going after these important questions.

Because the conclusions of the paper are focused on a single (albeit well validated) reporter in different types of motor neurons, it is hard to determine whether the mechanism of spared branch inhibition of regeneration requires wnd-puc (DLK/cJun) signaling, or whether this is a binary/threshold response in all contexts (for example, sensory axons or interneurons). However, the author points out in the response that there are sensory neuron examples where a spared connection does not block DLK activation. As such, it may not be a universal mechanism but could provide a model for better understanding of DLK control across different contexts.

Comments on revisions:

The new panels in Figure 1E do not have Y-axis labels. (mean puc-lacZ intensity?)

---

## [Author Response]

The following is the authors’ response to the original reviews.

**Reviewer #1 (Public review):**
This manuscript presents an interesting exploration of the potential activation mechanisms of DLK following axonal injury. While the experiments are beautifully conducted and the data are solid, I feel that there is insufficient evidence to fully support the conclusions made by the authors.In this manuscript, the authors exclusively use the puc-lacZ reporter to determine the activation of DLK. This reporter has been shown to be induced when DLK is activated.However, there is insufficient evidence to confirm that the absence of reporter activation necessarily indicates that DLK is inactive. As with many MAP kinase pathways, the DLK pathway can be locally or globally activated in neurons, and the level of DLK activation may depend on the strength of the stimulation. This reporter might only reflect strong DLK activation and may not be turned on if DLK is weakly activated. The results presented in this manuscript support this interpretation. Strong stimulation, such as axotomy of all synaptic branches, caused robust DLK activation, as indicated by puc-lacZ expression. In contrast, weak stimulation, such as axotomy of some synaptic branches, resulted in weaker DLK activation, which did not induce the puc-lacZ reporter. This suggests that the strength of DLK activation depends on the severity of the injury rather than the presence of intact synapses. Given that this is a central conclusion of the study, it may be worthwhile to confirm this further. Alternatively, the authors may consider refining their conclusion to better align with the evidence presented.

In Figure 1E we have replotted the *puc*-lacZ data to show comparisons between different injuries that leave different numbers of spared (or lost) boutons and branches. We observed no differences between injuries that remove only a small fraction of boutons (injury location (a)) and injuries that remove nearly all of them (injury locations (b) and (c)) and uninjured neurons (Figure 1E). These observations argue against the interpretation that the strength of DLK activation (at least within the cell body) depends on the severity of injury. Rather, *puc*-lacZ induction appears to be bimodal. It is either induced (in various injuries that remove all synaptic boutons), or not induced, including in injuries that spared only a small fraction of the total boutons. We therefore think that the presence of a remaining synaptic connection rather than the extent of the injury *per se* is a major determinant of whether the cell body component of Wnd signaling can be activated.

The reviewer (and others) fairly point out that our current study focuses on *puc*-lacZ as a reporter of Wnd signaling in the cell body. We consider this to be a downstream integration of events in axons that are more challenging to detect. It is striking that this integration appears strongly sensitized to the presence of spared synaptic boutons. Examination of Wnd’s activation in axons and synapses is a goal for our future work.

As noted by the authors, DLK has been implicated in both axon regeneration and degeneration. Following axotomy, DLK activation can lead to the degeneration of distal axons, where synapses are located. This raises an important question: how is DLK activated in distal axons? The authors might consider discussing the significance of this "synapse connection-dependent" DLK activation in the broader context of DLK function and activation mechanisms.

While it has been noted that inhibition of DLK can mildly delay Wallerian degeneration (Miller et al., 2009), this does not appear to be the case for retinal ganglion cell axons following optic nerve crush (Fernandes et al., 2014). It is also not the case for *Drosophila* motoneurons and NMJ terminals following peripheral nerve injury (Xiong et al., 2012; Xiong and Collins, 2012). Instead, overexpression of Wnd or activation of Wnd by a conditioning injury leads to an opposite phenotype - an increase in resiliency to Wallerian degeneration for axons that have been previously injured (Xiong et al., 2012; Xiong and Collins, 2012). The downstream outcome of Wnd activation is highly dependent on the context; it may be an integration of the outcomes of local Wnd/DLK activation in axons with downstream consequences of nuclear/cell body signaling. The current study suggests some rules for the cell body signaling, however, how Wnd is regulated at synapses and why it promotes degeneration in some circumstances but not others are important future questions.

For the reviewer’s suggestion, it is interesting to consider DLK’s potential contributions to the loss of NMJ synapses in a mouse model of ALS (Le Pichon et al., 2017; Wlaschin et al., 2023). Our findings suggest that the synaptic terminal is an important locus of DLK regulation, while dysfunction of NMJ terminals is an important feature of the ‘dying back’ hypothesis of disease etiology (Dadon-Nachum et al., 2011; Verma et al., 2022). We propose that the regulation of DLK at synaptic terminals is an important area for future study, and may reveal how DLK might be modulated to curtail disease progression. Of note, DLK inhibitors are in clinical trials (Katz et al., 2022; Le et al., 2023; Siu et al., 2018), but at least some have been paused due to safety concerns (Katz et al., 2022). Further understanding of the mechanisms that regulate DLK are needed to understand whether and how DLK and its downstream signaling can be tuned for therapeutic benefit.

**Reviewer #2 (Public review):**
Summary:The authors study a panel of sparsely labeled neuronal lines in Drosophila that each form multiple synapses. Critically, each axonal branch can be injured without affecting the others, allowing the authors to differentiate between injuries that affect all axonal branches versus those that do not, creating spared branches. Axonal injuries are known to cause Wnd (mammalian DLK)-dependent retrograde signals to the cell body, culminating in a transcriptional response. This work identifies a fascinating new phenomenon that this injury response is not all-or-none. If even a single branch remains uninjured, the injury signal is not activated in the cell body. The authors rule out that this could be due to changes in the abundance of Wnd (perhaps if incrementally activated at each injured branch) by Wnd, Hiw's known negative regulator. Thus there is both a yet-undiscovered mechanism to regulate Wnd signaling, and more broadly a mechanism by which the neuron can integrate the degree of injury it has sustained. It will now be important to tease apart the mechanism(s) of this fascinating phenomenon. But even absent a clear mechanism, this is a new biology that will inform the interpretation of injury signaling studies across species.Strengths:(1) A conceptually beautiful series of experiments that reveal a fascinating new phenomenon is described, with clear implications (as the authors discuss in their Discussion) for injury signaling in mammals.(2) Suggests a new mode of Wnd regulation, independent of Hiw.Weaknesses:(1) The use of a somatic transcriptional reporter for Wnd activity is powerful, however, the reporter indicates whether the transcriptional response was activated, not whether the injury signal was received. It remains possible that Wnd is still activated in the case of a spared branch, but that this activation is either local within the axons (impossible to determine in the absence of a local reporter) or that the retrograde signal was indeed generated but it was somehow insufficient to activate transcription when it entered the cell body. This is more of a mechanistic detail and should not detract from the overall importance of the study

We agree. The *puc*-lacZ reporter tells us about signaling in the cell body, but whether and how Wnd is regulated in axons and synaptic branches, which we think occurs upstream of the cell body response, remains to be addressed in future studies.

(2) That the protective effect of a spared branch is independent of Hiw, the known negative regulator of Wnd, is fascinating. But this leaves open a key question: what is the signal?

This is indeed an important future question, and would still be a question even if Hiw were part of the protective mechanism by the spared synaptic branch. Our current hypothesis (outlined in Figure 4) is that regulation of Wnd is tied to the retrograde trafficking of a signaling organelle in axons. The Hiw-independent regulation complements other observations in the literature that multiple pathways regulate Wnd/DLK (Collins et al., 2006; Feoktistov and Herman, 2016; Klinedinst et al., 2013; Li et al., 2017; Russo and DiAntonio, 2019; Valakh et al., 2013). It is logical for this critical stress response pathway to have multiple modes of regulation that may act in parallel to tune and restrain its activation.

**Reviewer #3 (Public review):**
Summary:This manuscript seeks to understand how nerve injury-induced signaling to the nucleus is influenced, and it establishes a new location where these principles can be studied. By identifying and mapping specific bifurcated neuronal innervations in the Drosophila larvae, and using laser axotomy to localize the injury, the authors find that sparing a branch of a complex muscular innervation is enough to impair Wallenda-puc (analogous to DLK-JNKcJun) signaling that is known to promote regeneration. It is only when all connections to the target are disconnected that cJun-transcriptional activation occurs.Overall, this is a thorough and well-performed investigation of the mechanism of sparedbranch influence on axon injury signaling. The findings on control of wnd are important because this is a very widely used injury signaling pathway across species and injury models. The authors present detailed and carefully executed experiments to support their conclusions. Their effort to identify the control mechanism is admirable and will be of aid to the field as they continue to try to understand how to promote better regeneration of axons.Strengths:The paper does a very comprehensive job of investigating this phenomenon at multiple locations and through both pinpoint laser injury as well as larger crush models. They identify a non-hiw based restraint mechanism of the wnd-puc signaling axis that presumably originates from the spared terminal. They also present a large list of tests they performed to identify the actual restraint mechanism from the spared branch, which has ruled out many of the most likely explanations. This is an extremely important set of information to report, to guide future investigators in this and other model organisms on mechanisms by which regeneration signaling is controlled (or not).Weaknesses:The weakest data presented by this manuscript is the study of the actual amounts of Wallenda protein in the axon. The authors argue that increased Wnd protein is being anterogradely delivered from the soma, but no support for this is given. Whether this change is due to transcription/translation, protein stability, transport, or other means is not investigated in this work. However, because this point is not central to the arguments in the paper, it is only a minor critique.

We agree and are glad that the reviewer considers this a minor critique; this is an area for future study. In Supplemental Figure 1 we present differences in the levels of an ectopically expressed GFP-Wnd-kinase-dead transgene, which is strikingly increased in axons that have received a full but not partial axotomy. We suspect this accumulation occurs downstream of the cell body response because of the timing. We observed the accumulations after 24 hours (Figure S1F) but not at early (1-4 hour) time points following axotomy (data not shown). Further study of the local regulation of Wnd protein and its kinase activity in axons is an important future direction.

As far as the scope of impact: because the conclusions of the paper are focused on a single (albeit well-validated) reporter in different types of motor neurons, it is hard to determine whether the mechanism of spared branch inhibition of regeneration requires wnd-puc (DLK/cJun) signaling in all contexts (for example, sensory axons or interneurons). Is the nerve-muscle connection the rule or the exception in terms of regeneration program activation?

DLK signaling is strongly activated in DRG sensory neurons following peripheral nerve injury (Shin et al., 2012), despite the fact that sensory neurons have bifurcated axons and their projections in the dorsal spinal cord are not directly damaged by injuries to the peripheral nerve. Therefore it is unlikely that protection by a spared synapse is a universal rule for all neuron types. However the molecular mechanisms that underlie this regulation may indeed be shared across different types of neurons but utilized in different ways. For instance, nerve growth factor withdrawal can lead to activation of DLK (Ghosh et al., 2011), however neurotrophins and their receptors are regulated and implemented differently in different cell types. We suspect that the restraint of Wnd signaling by the spared synaptic branch shares a common underlying mechanism with the restraint of DLK signaling by neurotrophin signaling. Further elucidation of the molecular mechanism is an important next step towards addressing this question.

Because changes in puc-lacZ intensity are the major readout, it would be helpful to better explain the significance of the amount of puc-lacZ in the nucleus with respect to the activation of regeneration. Is it known that scaling up the amount of puc-lacZ transcription scales functional responses (regeneration or others)? The alternative would be that only a small amount of puc-lacZ is sufficient to efficiently induce relevant pathways (threshold response).

While induction of *puc*-lacZ expression correlates with Wnd-mediated phenotypes, including sprouting of injured axons (Xiong et al., 2010), protection from Wallerian degeneration (Xiong et al., 2012; Xiong and Collins, 2012) and synaptic overgrowth (Collins et al., 2006), we have not observed any correlation between the degree of *puc*-lacZ induction (eg modest, medium or high) and the phenotypic outcomes (sprouting, overgrowth, etc). Rather, there appears to be a striking all-or-none difference in whether *puc*-lacZ is induced or not induced. There may indeed be a threshold that can be restrained through multiple mechanisms. We posit in figure 4 that restraint may take place in the cell body, where it can be influenced by the spared bifurcation.

**Recommendations for the authors:**

**Reviewer #2 (Recommendations for the authors):**
This is a beautiful study. Naturally, you're searching now for the underlying mechanism.A few questions:(1) At present you can not determine if the Wnd signal is never initiated (when a spared branch is present) or if it gets to the cell body but is incapable of activating the puckered reporter. Is there any optical reporter (JNK activation?) that could differentiate this?

The reviewer is correct that a tool to detect local activity of JNK kinase in axons would be ideal for probing the mechanisms that underlie our observations. A FRET reporter for JNK kinase activity has been developed and utilized in cultured cells (Fosbrink et al. 2010). It would be interesting to implement this reporter in *Drosophila*; it would need to be sensitive enough to visualize in single *Drosophila* axons. We have previously noted Wnd-dependent phosphorylated JNK in the cell body of injured motoneurons following nerve crush (Xiong et al., 2010). However anti-pJNK antibodies detect what appears to be a constitutive signal in uninjured axons that does not appear to be influenced by activation or inhibition of Wnd (Xiong et al., 2010).

(2) What happens when you injure the axon in a dSarm KO? This is more of a curiosity, not a necessity, but is it the axon dying or the detection of the injury itself?

We have tested whether overexpression of Nmnat or the *WldS* transgene, which inhibit Wallerian degeneration of injured axons, affect the induction of *puc*-lacZ following nerve injury. This manipulation has no effect on *puc*-lacZ expression in uninjured animals, and also has no effect on the induction of *puc*-lacZ following peripheral nerve crush (TJ Waller, personal communication).

(3) Are Wnd rescue experiments possible in this context? Would be an interesting place to do Wnd structure-function and compare it to the synaptic work.

This is not possible with current reagents. Expression of wild type *wnd* cDNA under the Gal4/UAS promoter leads to strong induction of *puc*-lacZ in uninjured animals, even when weak Gal4 driver lines are used (Xiong et al., 2012, 2010). Similar observations of constitutively active signaling have been observed for expression studies of DLK in mammalian cells ((Hao et al., 2016; Huntwork-Rodriguez et al., 2013; Nihalani et al., 2000), and data not shown). These and other observations suggest that the levels of Wnd/DLK protein are tightly controlled by posttranscriptional mechanisms. Delineation of sequences within Wnd/DLK that are required for its regulation would be helpful for addressing this question.

This will be required reading in my lab.

That is an honor. We look forward to help from the field to understand how and why this pathway is restrained at synapses. Your students may bring new ideas to the table.

**Reviewer #3 (Recommendations for the authors):**
Piezo is spelled incorrectly in the supplemental table in multiple places.

Thank you for pointing this out! We have made the correction.

References cited (in rebuttal)

Collins CA, Wairkar YP, Johnson SL, DiAntonio A. 2006. Highwire restrains synaptic growth by attenuating a MAP kinase signal. Neuron 51:57–69.

Dadon-Nachum M, Melamed E, Offen D. 2011. The “dying-back” phenomenon of motor neurons in ALS. J Mol Neurosci 43:470–477.

Feoktistov AI, Herman TG. 2016. Wallenda/DLK protein levels are temporally downregulated by Tramtrack69 to allow R7 growth cones to become stationary boutons. Development 143:2983–2993.

Fernandes KA, Harder JM, John SW, Shrager P, Libby RT. 2014. DLK-dependent signaling is important for somal but not axonal degeneration of retinal ganglion cells following axonal injury. Neurobiol Dis 69:108–116.

Ghosh AS, Wang B, Pozniak CD, Chen M, Watts RJ, Lewcock JW. 2011. DLK induces developmental neuronal degeneration via selective regulation of proapoptotic JNK activity. J Cell Biol 194:751–764.

Hao Y, Frey E, Yoon C, Wong H, Nestorovski D, Holzman LB, Giger RJ, DiAntonio A, Collins C. 2016. An evolutionarily conserved mechanism for cAMP elicited axonal regeneration involves direct activation of the dual leucine zipper kinase DLK. Elife 5. doi:10.7554/eLife.14048

Huntwork-Rodriguez S, Wang B, Watkins T, Ghosh AS, Pozniak CD, Bustos D, Newton K, Kirkpatrick DS, Lewcock JW. 2013. JNK-mediated phosphorylation of DLK suppresses its ubiquitination to promote neuronal apoptosis. J Cell Biol 202:747–763.

Katz JS, Rothstein JD, Cudkowicz ME, Genge A, Oskarsson B, Hains AB, Chen C, Galanter J, Burgess BL, Cho W, Kerchner GA, Yeh FL, Ghosh AS, Cheeti S, Brooks L, Honigberg L, Couch JA, Rothenberg ME, Brunstein F, Sharma KR, van den Berg L, Berry JD, Glass JD. 2022. A Phase 1 study of GDC-0134, a dual leucine zipper kinase inhibitor, in ALS. Ann Clin Transl Neurol 9:50–66.

Klinedinst S, Wang X, Xiong X, Haenfler JM, Collins CA. 2013. Independent pathways downstream of the Wnd/DLK MAPKKK regulate synaptic structure, axonal transport, and injury signaling. J Neurosci 33:12764–12778.

Le K, Soth MJ, Cross JB, Liu G, Ray WJ, Ma J, Goodwani SG, Acton PJ, Buggia-Prevot V, Akkermans O, Barker J, Conner ML, Jiang Y, Liu Z, McEwan P, Warner-Schmidt J, Xu A, Zebisch M, Heijnen CJ, Abrahams B, Jones P. 2023. Discovery of IACS-52825, a potent and selective DLK inhibitor for treatment of chemotherapy-induced peripheral neuropathy. J Med Chem 66:9954–9971.

Le Pichon CE, Meilandt WJ, Dominguez S, Solanoy H, Lin H, Ngu H, Gogineni A, Sengupta Ghosh A, Jiang Z, Lee S-H, Maloney J, Gandham VD, Pozniak CD, Wang B, Lee S, Siu M, Patel S, Modrusan Z, Liu X, Rudhard Y, Baca M, Gustafson A, Kaminker J, Carano RAD, Huang EJ, Foreman O, Weimer R, Scearce-Levie K, Lewcock JW. 2017. Loss of dual leucine zipper kinase signaling is protective in animal models of neurodegenerative disease. Sci Transl Med 9. doi:10.1126/scitranslmed.aag0394

Li J, Zhang YV, Asghari Adib E, Stanchev DT, Xiong X, Klinedinst S, Soppina P, Jahn TR, Hume RI, Rasse TM, Collins CA. 2017. Restraint of presynaptic protein levels by Wnd/DLK signaling mediates synaptic defects associated with the kinesin-3 motor Unc-104. Elife 6. doi:10.7554/eLife.24271

Miller BR, Press C, Daniels RW, Sasaki Y, Milbrandt J, DiAntonio A. 2009. A dual leucine kinase-dependent axon self-destruction program promotes Wallerian degeneration. Nat Neurosci 12:387–389.

Nihalani D, Merritt S, Holzman LB. 2000. Identification of structural and functional domains in mixed lineage kinase dual leucine zipper-bearing kinase required for complex formation and stress-activated protein kinase activation. J Biol Chem 275:7273–7279.

Russo A, DiAntonio A. 2019. Wnd/DLK is a critical target of FMRP responsible for neurodevelopmental and behavior defects in the Drosophila model of fragile X syndrome. Cell Rep 28:2581–2593.e5.

Shin JE, Cho Y, Beirowski B, Milbrandt J, Cavalli V, DiAntonio A. 2012. Dual leucine zipper kinase is required for retrograde injury signaling and axonal regeneration. Neuron 74:1015– 1022.

Siu M, Sengupta Ghosh A, Lewcock JW. 2018. Dual Leucine Zipper Kinase Inhibitors for the Treatment of Neurodegeneration. J Med Chem 61:8078–8087.

Valakh V, Walker LJ, Skeath JB, DiAntonio A. 2013. Loss of the spectraplakin short stop activates the DLK injury response pathway in Drosophila. J Neurosci 33:17863–17873.

Verma S, Khurana S, Vats A, Sahu B, Ganguly NK, Chakraborti P, Gourie-Devi M, Taneja V. 2022. Neuromuscular junction dysfunction in amyotrophic lateral sclerosis. Mol Neurobiol 59:1502–1527.

Wlaschin JJ, Donahue C, Gluski J, Osborne JF, Ramos LM, Silberberg H, Le Pichon CE. 2023. Promoting regeneration while blocking cell death preserves motor neuron function in a model of ALS. Brain 146:2016–2028.

Xiong X, Collins CA. 2012. A conditioning lesion protects axons from degeneration via the Wallenda/DLK MAP kinase signaling cascade. J Neurosci 32:610–615.

Xiong X, Hao Y, Sun K, Li J, Li X, Mishra B, Soppina P, Wu C, Hume RI, Collins CA. 2012. The Highwire ubiquitin ligase promotes axonal degeneration by tuning levels of Nmnat protein. PLoS Biol 10:e1001440.

Xiong X, Wang X, Ewanek R, Bhat P, Diantonio A, Collins CA. 2010. Protein turnover of the Wallenda/DLK kinase regulates a retrograde response to axonal injury. J Cell Biol 191:211– 223.